# Monitoring of Oil Spill Risk in Coastal Areas Based on Polarimetric SAR Satellite Images and Deep Learning Theory



**Lu Liao** [1,2]**, Qing Zhao** [1] **and Wenyue Song** [3,*]

1 School of Resources and Environment, University of Electronic Science and Technology of China, Chengdu 611731, China; liaolusc@uestc.edu.cn (L.L.); zhaoq@uestc.edu.cn (Q.Z.)
2 Technology Service Center of Surveying and Mapping, Sichuan Bureau of Surveying, Mapping and Geoinformation, Chengdu 610081, China
3 School of Resources and Environmental Engineering, Anhui University, Hefei 230601, China
* Correspondence: 16023@ahu.edu.cn

**Abstract:** Healthy coasts have a high ecological service value. However, many coastal areas are faced with oil spill risks. The Synthetic Aperture Radar (SAR) remote sensing technique has become an effective tool for monitoring the oil spill risk in coastal areas. In this study, taking Jiaozhou Bay in China as the study area, an innovative oil spill monitoring framework was established based on Polarimetric SAR (PolSAR) images and deep learning theory. Specifically, a DeepLabv3+-based semantic segmentation model was trained using 35 Sentinel-1 satellite images of oil films on the sea surface from maritime sectors in different regions all over the world, which not only considered the information from the PolSAR images but also meteorological conditions; then, the well-trained framework was deployed to identify the oil films in the Sentinel-1 images of Jiaozhou Bay from 2017 to 2019. The experimental results show that the detection accuracies of the proposed oil spill detection model were higher than 0.95. It was found that the oil films in Jiaozhou Bay were mainly concentrated in the vicinity of the waterways and coastal port terminals, that the occurrence frequency of oil spills in Jiaozhou Bay decreased from 2017 to 2019, and that more than 80 percent of the oil spill events occurred at night, mainly coming from the illegal discharge of waste oil from ships. These data indicate that, in the future, the PolSAR technique will play a more important role in oil spill monitoring for Jiaozhou Bay due to its capability to capture images at night.

**Keywords:** oil film; water pollution; remote sensing; coastal area; deep learning

## 1. Introduction

As a part of the ocean, coastal areas provide a large portion of the material and energy needs of human beings, and also provide a livable ecological environment for marine species [1]. Therefore, the ecological environment of coastal areas is of great significance to human beings. However, due to the uncontrolled exploitation of marine resources and discharge of pollutants, the ecological environments of many coastal areas are facing some ecological risks—typically, oil spill risks.

With the rapid development of coastal economies, more and more cargo ships are transporting goods at sea, which is bound to cause an increase in the occurrence of oil spill accidents [2,3]. Oil films spread and drift with ocean currents, causing large-scale marine pollution and fatal effects on marine ecology and lives [4]. Oil products can inhibit the photosynthesis of phytoplankton and reduce the dissolved oxygen in water, thus affecting the survival of marine organisms [5]. They can also cause deformations in hatched fish, and the heavy metal elements from oil spills accumulate in the body of the fish, which in turn endangers human health through consumption [6]. It has been reported that, since the 1970s, over five million tons of oil have been released into marine environments due to accidents [7]. For example, in 2018, the ship Sanchi departed from Panama and then collided with a bulk carrier vessel, which departed from Hong Kong. This accident caused

oil films that covered more than 100 km$^2$ of the sea's surface [8], and it took about two weeks to complete the initial clean-up of the oil spill.

To take effective actions to protect the environment of coastal areas, oil spill risk monitoring studies are needed. Early studies mainly relied on historical statistical data or on data sampled from in-situ stations [9]. It is essential to implement spatially continuous monitoring of the sea surface to obtain a more comprehensive understanding of the distribution of risk sources. The satellite remote sensing technique is capable of capturing large-scale images of the Earth, and has been utilized in the study of the ecological risk monitoring of coastal areas [10–13]. Researchers have presented a number of methods to identify marine oil spills from remote sensing images [14–16]. Compared with optical remote sensing, which is easily affected by both clouds and the solar flare effect occurring on the sea surface, Synthetic Aperture Radar (SAR) has unique advantages thanks to its all-weather and all-condition imaging capabilities. Besides this, oil films can weaken the scattering signals from the water, making oil films exhibit a dark-spot appearance in SAR images that differs from the surrounding water in most cases. This makes SAR data an important data source for detecting marine oil spills [16]. Therefore, a lot of studies have focused on oil spill detection from SAR images [17–21].

The study in [17] revealed that texture analysis is an efficient approach for detecting oil spill areas in SAR images, and they proposed a novel oil spill detection approach using texture analysis and an artificial neural network. Skrunes et al. [18] used multi-source SAR images to validate the capability of polarimetric features, such as polarimetric decomposition parameters and the copolarized phase difference, in identifying oil types. A recent work by Dong et al. [19] presented a semi-automatic oil spill detection approach from Sentinel-1 images by combining manual annotation and computer-aided interpretation. In recent years, due to their good performance in representation learning and in mining useful information from big data, deep learning-based methods have been applied to detect oil spills in SAR images [20]. Chen et al. [21] applied two different deep learning-based detection models to select the optimal polarimetric feature combinations from quad-polarimetric SAR images, which showed better performances than traditional supervised classification methods; Shaban et al. [15] utilized a multi-layer convolutional neural network (CNN) for oil spill classification and semantic segmentation from SAR images, as did a more recent study conducted by Ma et al. [16].

Although more and more studies are focusing on oil spill identification from PolSAR images via deep learning-based methods, some bottlenecks still need to be solved—typically, the "look-alikes" phenomenon. Look-alikes refer to targets on the sea surface that also have weak backscattering signals similar to those of oil films, such as the wake caused by ships, heavy rainfall regions, and low wind-speed regions [16]. To distinguish oil films from these look-alikes is challenging work, which means that more information should be considered when training a deep learning-based oil spill detection model. Based on this, this paper presents a DeepLabv3+-based semantic segmentation model for the task of oil spill detection from PolSAR satellite images. Different from most related studies that have only utilized the backscattering intensity signal data, the proposed model considers more information, including more polarimetric features and some information on meteorological conditions. We first trained the model with 35 Sentinel-1 satellite images of oil films on the sea surface from maritime sectors in different regions all over the world, and then used the well-trained model to identify and assess the oil spill risk in Jiaozhou Bay from 2017 to 2019.

The study objectives of this paper are to develop a reliable and robust algorithm to identify oil films on the ocean from SAR images, and to apply the identification results to assess the oil spill risk in the coasts. We summarize the main contributions of our work as follows: (1) More information from both the PolSAR dataset and meteorological dataset are taken into account by the oil spill monitoring framework, which can significantly alleviate the influence of look-alikes; (2) Sufficient images with oil films on the sea surface are input into the training step of the proposed model, which ensures the robustness of the model;

(3) The oil spill risk in Jiaozhou Bay over multiple years is comprehensively monitored and evaluated.

The rest of this paper is organized as follows: Section 2 introduces some basic information about Jiaozhou Bay and the datasets used in this study; Section 3 describes the proposed oil spill detection method; then, the experimental results and the discussion are provided in Section 4; finally, conclusions are drawn in Section 6.

## 2. Study Area and Data Sources

### 2.1. Study Area

Jiaozhou Bay is located in the Yellow Sea near the south coast of the Shandong Peninsula and is a semi-enclosed port of Qingdao (Figure 1). It covers an area of approximately 416 km$^2$ and serves as a vital hub for maritime trade in China. The study area is located in the warm, temperate monsoon climate zone, and is mainly influenced by southeasterly winds in the summer and northerly winds in the winter, with an average yearly rainfall of 600–800 mm [22].

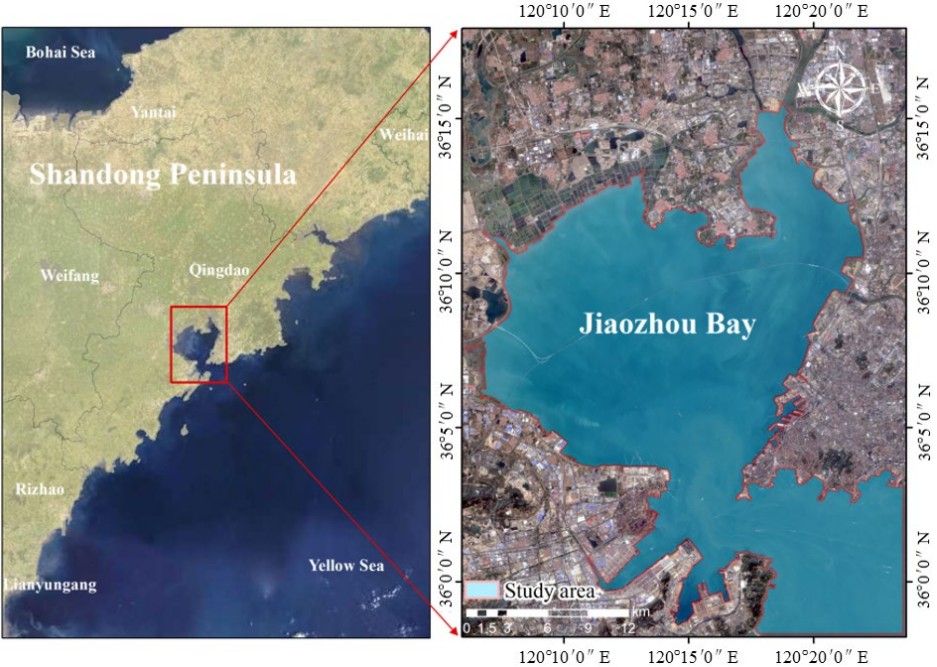

**Figure 1.** Map of the study area.

Jiaozhou Bay is known as Qingdao's "mother bay" [23]. There are a number of aquaculture industries located in the shallow eastern, western, and northern waters of the bay. The tourism industry in the surrounding urban areas is booming, and offshore marine trades have rapidly developed over the last decade. A direct threat to the developed marine trades is the release of oil. It must be pointed out that, due to the negative influence of the COVID-19 pandemic on marine trade activities around China, this study only focuses on the monitoring of oil spills in Jiaozhou Bay from 2017 to 2019, i.e., before the COVID-19 pandemic.

### 2.2. Data Sources

In this work, Sentinel-1 satellite images were used for oil spill detection. Sentinel-1 is a C-band SAR system composed of two satellites. Sentinel-1 satellite data can be obtained from the European Space Agency (ESA) website (https://scihub.copernicus.eu, accessed on 11 March 2023). Sentinel-1 has a variety of imaging modes, working in single polarization and dual polarization modes [24,25]. Radar waves are susceptible to the physical traits and structures of the target, which ensures the good capability of imaging radar systems to obtain the scattering information of the target.

The revisit period of Sentinel-1 satellites is 6 days; that is to say, users can obtain the Sentinel-1 image of a same area every 6 days. In many cases, the cleanup process for oil films on the ocean lasts more than a week. This means that the observation frequency and image number of the Sentinel-1 system is sufficient to ensure the reliability of the observed phenomena in this study. The interferometric wide swath (IW) imaging mode of Sentinel-1 has the advantages of a wide coverage and high resolution (5 m × 20 m), which makes it suitable for oil spill detection on the sea surface [26]. In this study, the dual-polarimetric SAR images acquired in Sentinel-1 IW mode were utilized to detect oil spills in Jiaozhou Bay. Monostatic dual-polarimetric data can be completely described by a 2 × 2 polarimetric covariance matrix [27,28]. Under the assumption of scattering reciprocity, the polarimetric covariance matrix of Sentinel-1 SAR data can be defined as [29]:

$$C = \begin{bmatrix} C_{11} & C_{12} \\ C_{21} & C_{22} \end{bmatrix} = \begin{bmatrix} 2|S_{VH}|^2 & \sqrt{2}S_{VH}S_{VV}^* \\ \sqrt{2}S_{VV}S_{VH}^* & |S_{VV}|^2 \end{bmatrix} \tag{1}$$

with

$$S_{VV} = |S_{VV}|^{e^{j\varnothing VV}} \tag{2}$$

$$S_{VH} = |S_{VH}|^{e^{j\varnothing VH}} \tag{3}$$

where $S_{VH}$ represents the scattering component of the horizontal transmission and vertical reception; $|S_{VH}|$ and $\varnothing_{VH}$ denote the amplitude and the phase of the VH polarization, respectively; and the other variables are similarly defined. * represents the conjugate operator and *j* is the imaginary unit.

In this study, a deep learning-based method was developed to detect oil spills from Sentinel-1 images acquired in the area of Jiaozhou Bay from 2017 to 2019. To train the network, we collected 35 Sentinel-1 images of oil films on the sea surface by obtaining oil spill event information from maritime sectors in different regions. The training dataset covered 17 regions around the world (Figure 2), mainly the Persian Gulf, the North Sea, the Red Sea, and the Mediterranean Sea. In both the training and detection steps, the Sentinel-1 images were preprocessed before being input into the network, including geocoding and speckle filtering [30].

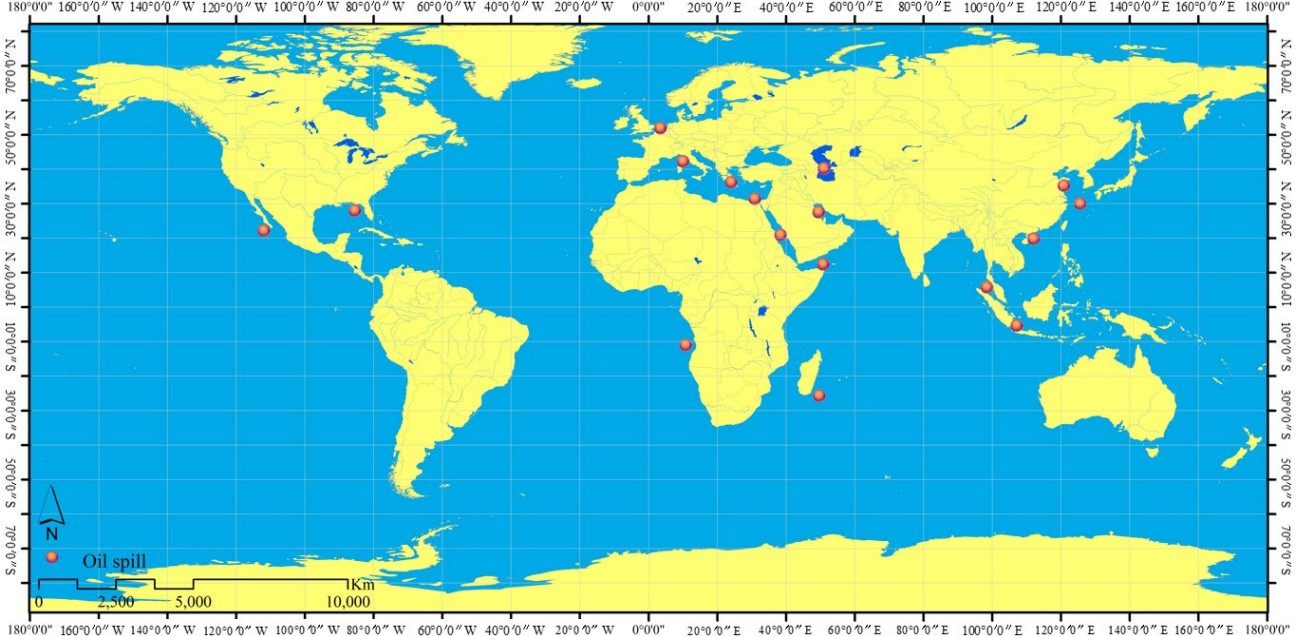

**Figure 2.** Oil spill location information for the model training dataset.

As per some other studies [31,32], weather conditions—such as wind speed and rainfall—can also change the Bragg scattering effect generated by the sea surface, thus hindering the oil spill detection approach from SAR images. Considering the above issue, we also took wind speed and rainfall information as features to train the detection networks. The wind speed information was obtained from the corresponding Sentinel-1 imagery based on a C-band geophysical model function (CMOD5) [33], which was implemented using SNAP software. The rainfall information was obtained through the Global Precipitation Measurement (GPM) rainfall data provided by NASA (https://disc.gsfc.nasa.gov, accessed on 11 March 2023).

## 3. Method

### 3.1. The Baic Architecture

In this study, a DeepLabv3+-based semantic segmentation algorithm was used to identify oil films in Jiaozhou Bay. Among all the semantic segmentation algorithms, the DeepLabv3+ framework [34] is one of the most effective ones, which improves on the DeepLabv3 framework [35] by adding a decoder to refine the segmentation results. The basic architecture of the DeepLabv3+ network model for oil spill detection is shown in Figure 3.

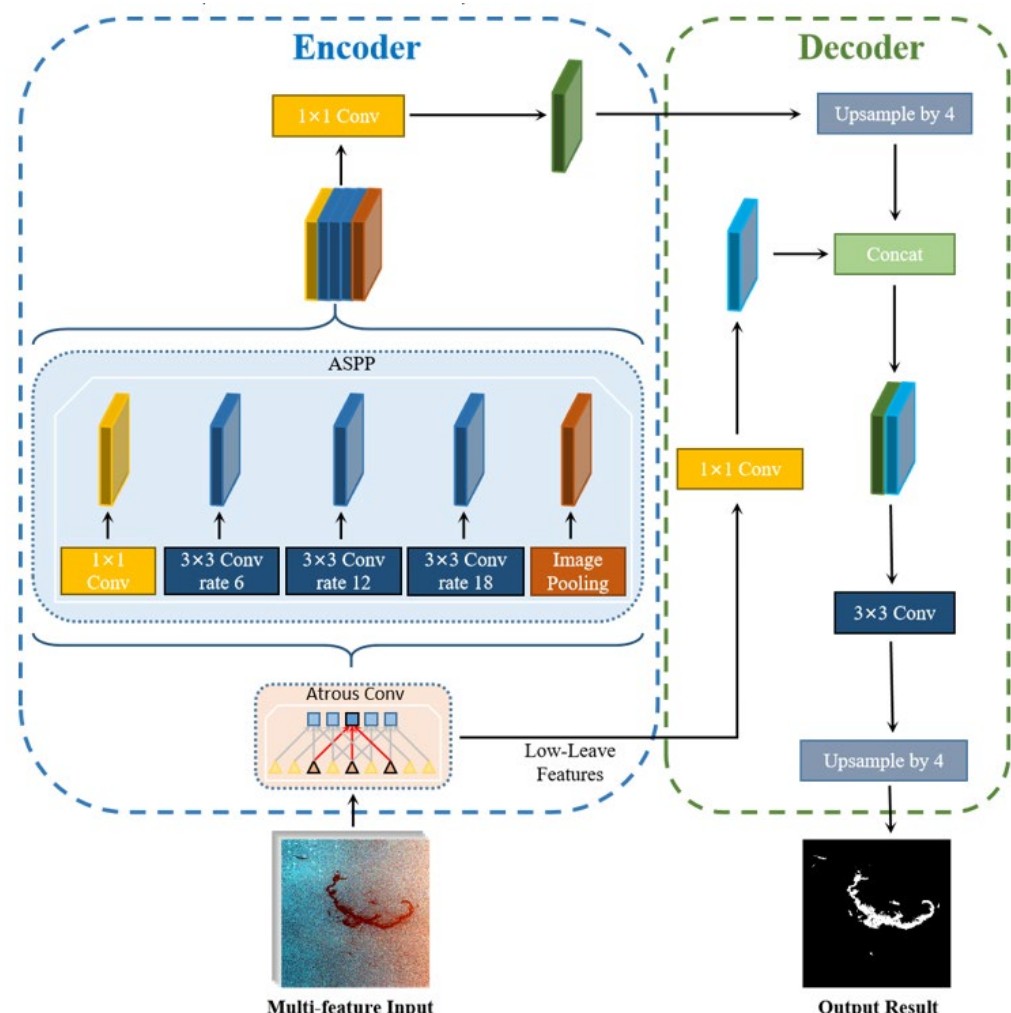

**Figure 3.** The basic architecture of the DeepLabv3 + network model for oil spill detection.

The encoder part consists of a DCNN with dilated convolution and an atrous spatial pyramid pooling (ASPP) module, where group normalization (GN) is introduced to eliminate the influence of batch size on accuracy. The encoder part introduces dilated

convolution to expand the receptive field and uses different receptive fields to extract multi-scale features so as to capture rich oil spill features, which are crucial to the accuracy of the segmentation [34].

The decoder part accepts the output of the low-level feature map from the middle layer of the backbone network and the ASPP module to achieve the purpose of restoring the target boundary details, thereby obtaining high-precision oil spill detection results.

*3.2. Multi-Feature Input*

Normally, oil films manifest themselves in SAR images as dark-spot appearances that differ from the surrounding water. However, "look-alikes"—such as the wake caused by ships, heavy rainfall regions, and low wind speed regions—can also lead to weak backscattering signals [16], which hinders the accurate identification of oil spills. Therefore, integrating more features into the oil spill detection model is beneficial for improving the detection accuracy.

For the Sentinel-1 PolSAR data, the amplitude and phase information (as shown in Equation (1)) were considered in this study. In addition, the polarimetric decomposition parameters were also considered. The polarimetric decomposition process is able to describe the physical scattering mechanisms of the targets well. In this stage, the Cloude–Pottier decomposition theory [36] was used to obtain the polarimetric decomposition parameters, i.e., the polarimetric entropy ($H$), average scattering angle ($\alpha$), and polarimetric anisotropy ($A$).

In the $H/A/\alpha$ decomposition model, $H$ represents the pureness of the scattering traits, which is:

$$H = -\sum_{\tau=1}^{2} p_\tau log_2 p_\tau, \ H \in [0,1] \tag{4}$$

with

$$p_\tau = \frac{\lambda_1}{\lambda_1 + \lambda_2} \tag{5}$$

where $\lambda_1$ and $\lambda_2$ are the two real eigenvalues of the polarimetric covariance matrix, as shown in Equation (1). A higher value of $H$ reveals that the contribution of the main scattering component of the target is higher.

$\alpha$ is employed to distinguish the dominant scattering component, and is defined as follows:

$$\alpha = \sum_{\tau=1}^{2} p_\tau cos^{-1}(|e_\tau(1)|), \ \alpha \in [0°, \ 90°] \tag{6}$$

An $\alpha$ value below 45° indicates that the dominant scattering mechanism of the target is odd scattering. As the value of $\alpha$ increases, the dominant scattering mechanism is more likely to be the double-bounce scattering.

$A$ quantifies the significance of the first eigenvalue with regard to the second eigenvalue:

$$A = \frac{(\lambda_1 - \lambda_2)}{(\lambda_1 + \lambda_2)}, \ A \in [0,1] \tag{7}$$

Considering that some natural factors such as low wind speed and heavy rainfall can hinder the identification of oil films, information on the ocean environment (including wind speed and rainfall) was also input into the network to further improve its performance in identifying oil spills. The wind speed information was obtained from the CMOD5 algorithm developed for the C-band radar [33]. In CMOD5, the normalized radar cross section $\sigma_{VV}^0$ is assumed to be related to the incidence angle $\varnothing$ as:

$$\sigma_{VV}^0 = b_1[1 + b_2 cos(\varnothing) + b_3 cos(2\varnothing)]^{1.6} \tag{8}$$

By inputting the $\sigma_{VV}^0$ and $\varnothing$ values of the ocean area in the Sentinel-1 images, the values of the variables $b_1$, $b_2$, and $b_3$ for each pixel of the ocean area can be calculated using

least-squares estimation. In addition, $b_1$, $b_2$, and $b_3$ have nonlinear relationships with the wind speed $D$:

$$b_1 = f_1(D) \tag{9}$$

$$b_2 = f_2(D) \tag{10}$$

$$b_3 = f_3(D) \tag{11}$$

The wind speed value of each pixel of the ocean area can be obtained using least-squares estimation.

The GPM rainfall dataset used in this study was obtained based on the GPM inversion algorithm proposed by [37]. The dataset provides global rainfall data with high spatial and temporal resolutions.

## 4. Experimental Results

### 4.1. Quantitative Assessment Indices

In this paper, two indicators were used to validate the oil spill identification performance—namely, the overall accuracy (*OA*) and mean intersection over union (*MIoU*) [28].

*OA* describes the ratio of the number of pixels correctly classified to the total number of pixels:

$$OA = \frac{TP + TN}{TP + TN + FP + FN} \tag{12}$$

where *TP* is true positive, which denotes the pixel number correctly predicted as oil spill; *TN* is true negative, which denotes the pixel number correctly predicted as background; *FP* is false positive, which denotes the pixel number in the background wrongly predicted as oil spill; and *FN* is false negative, which denotes the pixel number in the oil spill wrongly predicted as background.

*MIoU* measures the average of the ratio of the intersection and the union of the oil spill and background, which is defined as:

$$MIoU = \frac{1}{2}\left(\frac{TN}{FN + TN + FP} + \frac{TP}{FN + TP + FP}\right) \tag{13}$$

### 4.2. Assessment of Oil Spill Identification Accuracy

In the proposed oil spill detection method, multiple features including the amplitude and phase of the polarimetric data, the polarimetric decomposition parameters (i.e., H/A/$\alpha$), and the meteorological conditions (i.e., wind speed and rainfall) were input into the DeepLabv3+ network. To validate the necessity of inputting multiple features for oil spill detection, we constructed three types of feature sets—namely, $\delta_1$, only considering amplitude and phase (see Equation (14)); $\delta_2$, considering amplitude, phase, and polarimetric decomposition parameters (see Equation (15)); and $\delta_3$, considering all the features mentioned above (see Equation (16)). Besides this, the oil spill detection results obtained by the traditional PolSAR Wishart supervised Classification method [38] are also reported for comparison.

$$\delta_1 = [|S_{HV}|, |S_{VV}|, \varnothing_{VH} - \varnothing_{VV}] \tag{14}$$

$$\delta 2 = [|S_{HV}|, |S_{VV}|, \varnothing_{VH} - \varnothing_{VV}, H, A, \alpha] \tag{15}$$

$$\delta_3 = [|S_{HV}|, |S_{VV}|, \varnothing_{VH} - \varnothing_{VV}, H, A, \alpha, D, R] \tag{16}$$

The oil spill detection results for some of the Sentinel-1 images are shown in Figure 4. We also list the quantitative assessment values in Table 1. As can be seen, many spots were exhibited in the identification results of the traditional PolSAR supervised classification method, which seriously degrades the identification accuracy. This is because the traditional

PolSAR supervised algorithms only take into account the shallow characteristics of the target while ignoring contextual information, and hence cannot provide fine segmentation maps. By considering the rich polarimetric and meteorological data, the detection precision of the proposed method was promoted. In particular—as marked in Figure 4—some look-alikes, which were misclassified as oil films by the other methods, were correctly classified by considering more information. Furthermore, the detected oil films were uniform, which is in line with their real state.

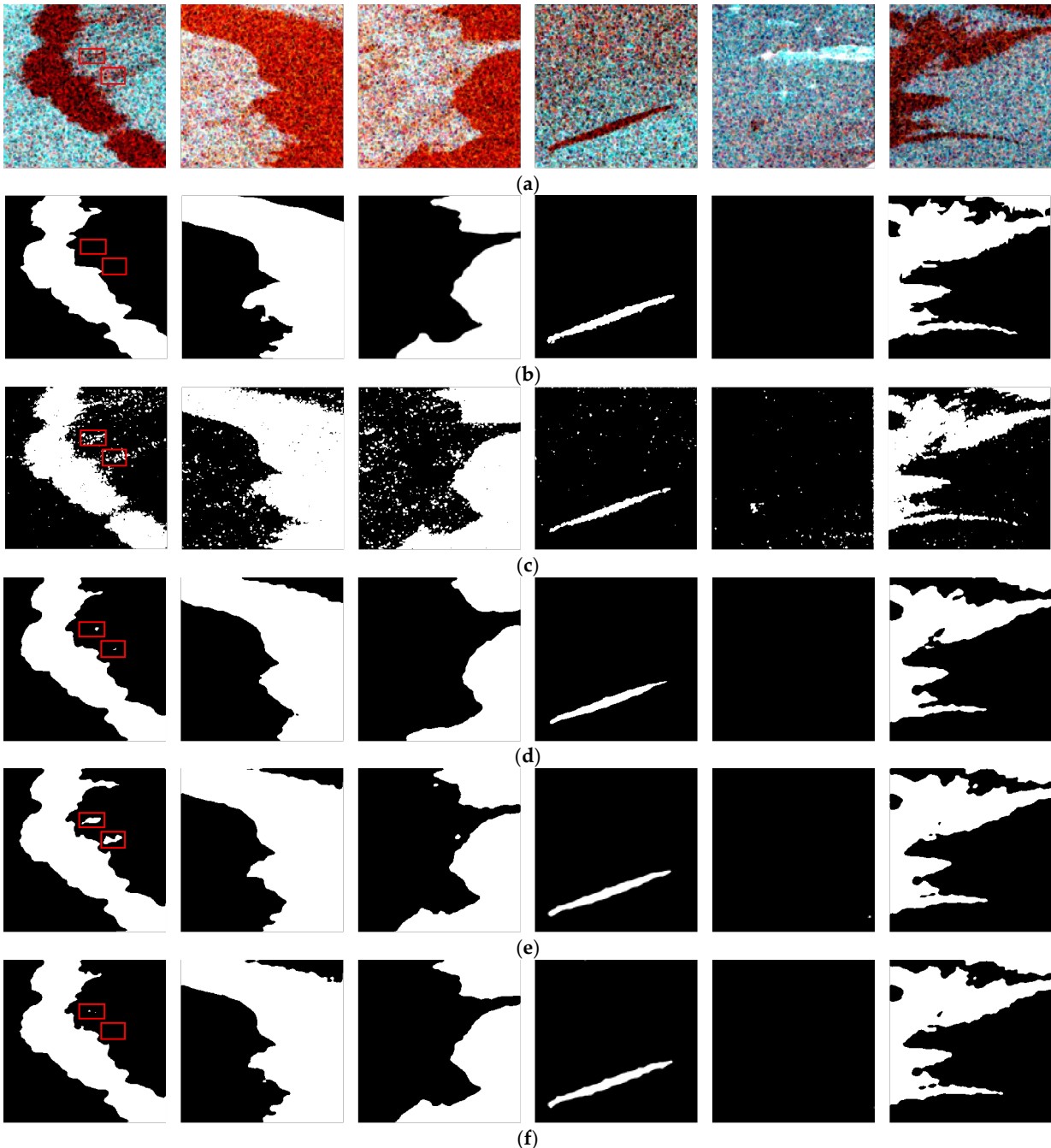

**Figure 4.** Oil spill detection results. (**a**) The original Sentinel-1 SAR images. (**b**) Ground-truth maps. (**c**) Detection results obtained by the PolSAR Wishart supervised classification method. (**d**) Detection results obtained by the proposed model using feature set $\delta_1$. (**e**) Detection results obtained by the proposed model using feature set $\delta_2$. (**f**) Detection results obtained by the proposed model using feature set $\delta_3$.

**Table 1.** Quantitative assessment values for different oil spill identification methods.

| Method | OA | MIoU |
|---|---|---|
| PolSAR Wishart | 0.8215 | 0.7238 |
| The proposed model using feature set $\delta_1$ | 0.9664 | 0.9180 |
| The proposed model using feature set $\delta_2$ | 0.9818 | 0.9559 |
| The proposed model using feature set $\delta_3$ | 0.9838 | 0.9606 |

To further validate the generalization capability of the proposed method in different marine areas, we also display the detection results for an image with oil films on the coast of Brazil in August 2019 (Figure 5). It can be observed that, in this new area, the detection result was still quite close to the label image. The OA and MIoU values were 0.9812 and 0.9599, respectively.

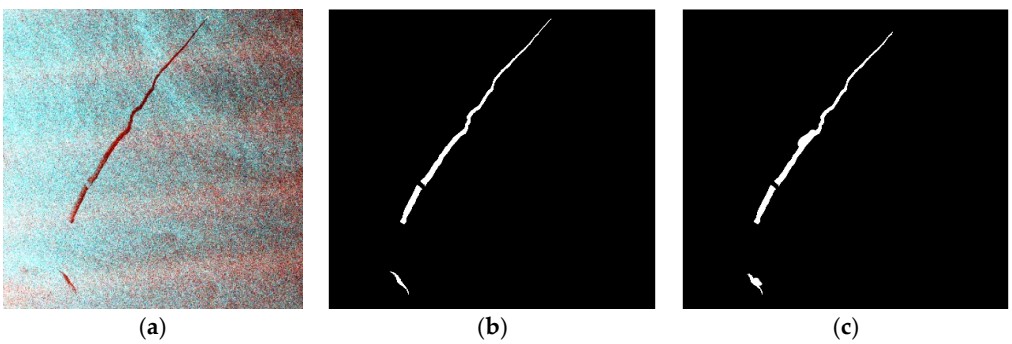

|  (**a**)  |  (**b**)  |  (**c**)  |

**Figure 5.** Oil spill detection results for the coast of Brazil in August 2019. (**a**) The original Sentinel-1 SAR images. (**b**) The label image. (**c**) Detection result obtained by the proposed method.

### 4.3. Monitoring of Oil Spill Risk in Jiaozhou Bay

We applied the above well-trained oil spill detection model to identify oil films in 245 Sentinel-1 SAR images of the Jiaozhou Bay area from 2017 to 2019. The number of images in each month for these three years is shown in Figure 6, and the number of images in which oil films were detected is also shown.

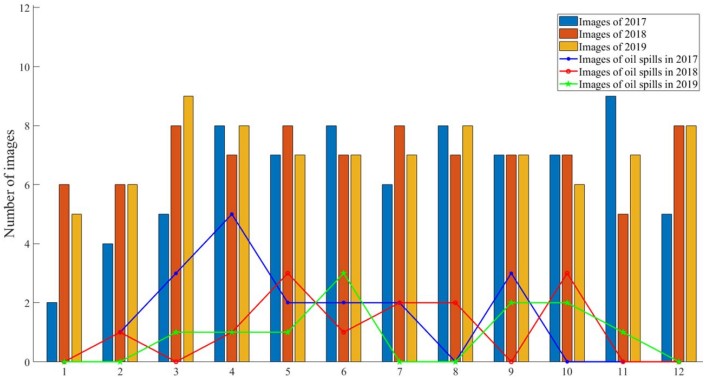

**Figure 6.** The number of oil spill events detected in the 245 SAR images from 2017 to 2019.

It can be seen that oil spill events were detected in most months. An interesting phenomenon is that, in winter, the occurrence frequency of oil spill events was notably lower than that in other seasons. The reason for this is that Jiaozhou Bay is prone to seawater icing in winter, due to the influence of cold air and low temperatures, and hence the activities of the marine trade and fishing industry are limited [39].

Some statistics concerning the oil films in Jiaozhou Bay are displayed in Figure 7. First of all, the proportion of oil spill events showed a downward trend from 42.86% in 2017 to 26.19% in 2019. This suggests that the marine management measures implemented in

Jiaozhou Bay over the last few years have effectively controlled the illegal release of oil. As pointed out in [40], a main control measure taken by the Qingdao Maritime Bureau is to punish ship owners who have illegally released oil by comparing the chemical compositions of the oil spills with that in the ships which were transported through the bay.

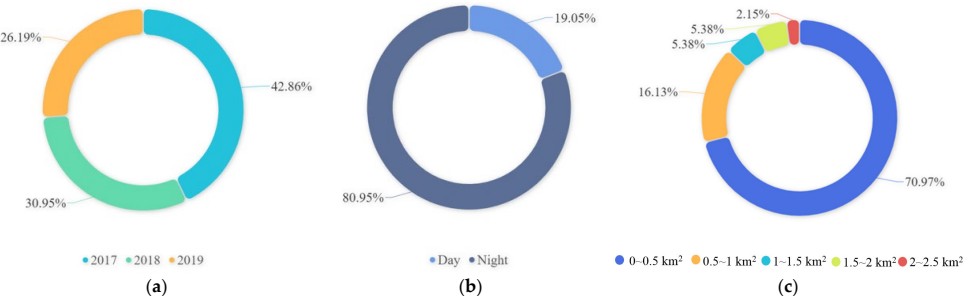

**Figure 7.** Statistics concerning the oil films observed in Jiaozhou Bay. (**a**) The proportion of oil spill events occurring each year. (**b**) The proportions of oil spill events in daytime and at night. (**c**) The proportions of oil spills of different sizes.

It can also be seen that oil spill events mainly occurred at night, accounting for 80.95% of the total. The reason for this is that, in Jiaozhou Bay, oil spills most likely come from the illegal cleaning of the oil tanks of ships at night [13]. The area statistics for the oil spills show that most of the oil spills were of a small size, further indicating that the oil spills in Jiaozhou Bay have mostly come from the illegal discharge of waste oil rather than from accidental oil spills or collision accidents.

The oil spill risk distribution maps for Jiaozhou Bay from 2017 to 2019 are displayed in Figure 8. It can be clearly seen that, in general, the frequency of oil spill events decreased year by year. Compared with 2017, the frequency of oil spill events occurring in the waterways and ports of Jiaozhou Bay in 2018 and 2019 significantly reduced, and the frequency of oil spill events occurring at the entrance of the bay in 2019 also significantly reduced, compared with previous years. In all three years, the locations with a high frequency of oil spill events were mainly concentrated in the vicinity of the waterways and coastal port terminals. Besides this, we can also observe that between different years, the highest areas of oil spill incidents were exhibited in different locations. According to the above observations and considering the fact that Jiaozhou Bay is one of the most important shipping ports of China, a conclusion can be drawn here is that the oil spills in the waters of Jiaozhou Bay are not caused by some fixed pollution points, and instead are most likely to be related to shipping activities.

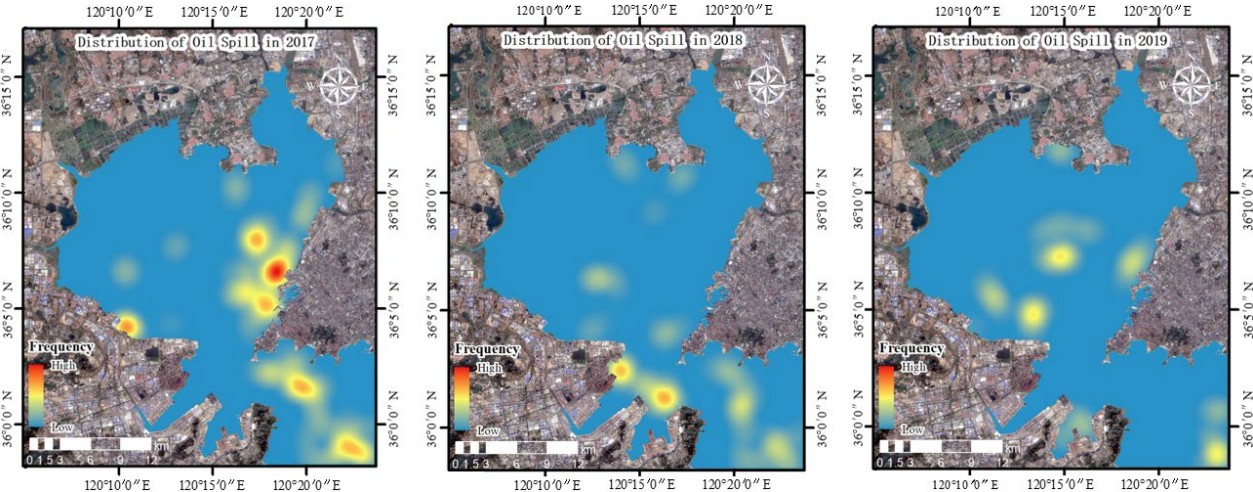

**Figure 8.** Distribution map of oil spill risk in Jiaozhou Bay from 2017 to 2019.

## 5. Discussion

### 5.1. Data Limitation

The SAR images used in this study were acquired by the Sentinel-1 system. Sentinel-1 consists of two satellites—namely, Sentinel-1 A and Sentinel-1 B. As introduced in Section 2.2, the revisit period of Sentinel-1 is 6 days. However, since 2022, the images of Sentinel-1 B have become unavailable due to a system breakdown. That is to say, after 2022, the revisit period of the Sentinel-1 system became 12 days. The long revisit period means that some oil films with a small area may not be observed by the satellites, because the cleanup period or natural disappearance period of those oil films are very likely to be shorter than the revisit period of the satellites. In such cases, some illegally released oil by ships in the coasts might not be observed by the Sentinel-1 system, since they often have a small area and are more quickly cleaned up. The aforementioned limitation implies that, in the future, using more data sources from different satellites is an effective means of ensuring a high observation frequency for the monitoring of oil spill events on the coasts.

### 5.2. Comparison with Related Works

To support the phenomena and conclusions we observed in Section 4, we compared our work with a previous study by Ma et al. [13]. In this work, the authors used remote sensing images to identify the oil spill events that occurred in Jiaozhou Bay from 2015 to 2021. The temporal distribution characteristics of the oil spills in Jiaozhou Bay found in [13] are displayed in Figure 9. Clearly, the study in [13] supports the observed phenomena in this paper that oil spill incidents mainly occur at night and that they occur at the lowest frequency in winter.

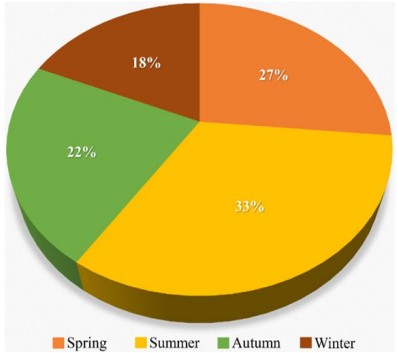 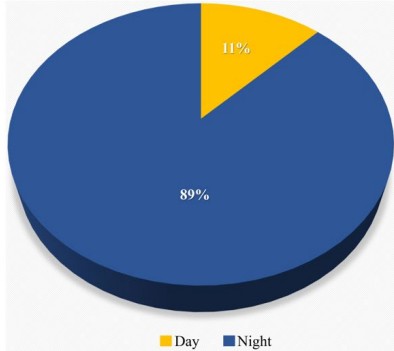

**Figure 9.** Seasonal, day, and night regularity of oil spills in Jiaozhou Bay reported by Ma et al. [13].

## 6. Conclusions

Coastal areas provide irreplaceable ecological value and represent a significant economic resource for human beings. Taking effective actions to protect the environment of coastal areas is of great significance. For this purpose, monitoring the oil spill risk in coastal areas and conducting risk assessments are important to understand the ecological situation of coastal areas and can assist with subsequent environmental protection plans. Some studies have been conducted to use SAR satellite data and deep learning theory to monitor the oil spill risk of coastal areas, but the monitoring processes and results are not reliable enough in most cases—mainly influenced by the appearance of look-alikes and the limited training dataset.

To solve the bottlenecks of the current oil spill detection methods from SAR images, this work presents an innovative oil spill risk-monitoring framework based on Sentinel-1 PolSAR satellite images and deep learning theory. A DeepLabv3+-based semantic segmentation model was trained using 35 Sentinel-1 images of oil films on the sea surface from maritime sectors in different regions all over the world, which not only considered the information from PolSAR images but also meteorological conditions; then, the well-trained framework was deployed to identify oil films in Sentinel-1 images of Jiaozhou Bay from

2017 to 2019. The qualitative and quantitative assessment results showed that the proposed oil spill detection method has a high detection accuracy; in the meantime, the detected oil films were uniform in the classification maps. The oil spill risk monitoring results of Jiaozhou Bay show that the oil films were mainly concentrated in the vicinity of the waterways and coastal port terminals, that the occurrence frequency of oil spills in the bay decreased from 2017 to 2019, and that most of the oil spills came from the illegal discharge of waste oil from ships at night.

**Author Contributions:** Conceptualization, L.L. and Q.Z.; methodology, L.L.; software, Q.Z.; validation, Q.Z.; formal analysis, Q.Z.; investigation, L.L.; resources, L.L.; data curation, L.L.; writing—original draft preparation, L.L. and W.S.; writing—review and editing, Q.Z.; visualization, L.L.; supervision, L.L.; project administration, L.L.; funding acquisition, W.S. All authors have read and agreed to the published version of the manuscript.

**Funding:** This work was supported by the Sichuan Science and Technology Program (No. 2023YFG0123, No. 2023YFS0381), and the Sichuan Bureau of Surveying, Mapping and Geoinformation Research Project on New Fundamental Surveying and Mapping Technologies (No. 2023KJ003).

**Institutional Review Board Statement:** Not applicable.

**Informed Consent Statement:** Not applicable.

**Data Availability Statement:** Not applicable.

**Conflicts of Interest:** The authors declare no conflict of interest.

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
