# Peer review of "Monitoring of Oil Spill Risk in Coastal Areas Based on Polarimetric SAR Satellite Images and Deep Learning Theory"

_sustainability, doi:10.3390/su151914504_

Round 1

Reviewer 1 Report

1.      Abstract means the results in summarized form, but here no results have been presented. But data and methodology are discussed in detail. Please rewrite the abstract and salient quantitive results may be presented in the abstract along with brief introductory sentences and future recommendations.

2.      What research gaps/limitations have the authors covered in their work?

3.      Add a data limitation section in the methodology section.

4.      Add the Discussion section which should compare observed phenomena with other studies related to this problem and provide a global or regional perspective of study outputs.

1.      Please thoroughly review the manuscript for its grammar.

Reviewer 2 Report

The manuscript needs to improve, as shown below:

1.       Clear objectives should be indicated at the end of the introduction.

2.       Page 9, line 296: The authors observed that the number of oil spill incidents occurred in most months of the year, but they concluded that they were lowest in the winter during the study period from 2017-2019 due to the decrease in trade activity. These findings need to be confirmed and supported by references.

3.       Page 9, line 305: The authors observed the proportion of oil spill events showed a downward trend from 42.86% in 2017 to 26.19% in 2019, which was related to the effectively controlled illegal release of oil. However, the author must explain the measures that contributed to this decline and support this with references.

4.       The authors must clarify that the total image number is sufficient for a good conclusion.

5.       Page 10, line 330: More explanations are needed to clarify these conclusions: “Some fixed pollution points do not cause the oil spills in the waters of Jiaozhou Bay, and instead are most likely related to shipping activities.”

6.       If possible, the researcher had to compare the current study results with other studies in similar areas.

non

Reviewer 3 Report

Dear authors,

The study is quite interesting and innovative.

I suggest repeating this study in other locations to see if it is possible to obtain similar results.

One tip would be to apply this study, if possible, to the crude oil spill accident that occurred on the coast of Brazil in August 2019.

One detail that caught my attention: the number of times the expression oil spill was repeated, especially in the second paragraph. Authors should be careful with these repetitions in a row.
